# Smokers’ and Non-Smokers’ Attitudes towards Smoking Cessation in Saudi Arabia: A Systematic Review

**DOI:** 10.3390/ijerph17218194

**Published:** 2020-11-06

**Authors:** Mansour Tobaiqy, Dennis Thomas, Andrew MacLure, Katie MacLure

**Affiliations:** 1Department of Pharmacology, College of Medicine, University of Jeddah, P.O. Box 45311, Jeddah 21512, Saudi Arabia; 2Priority Research Centre for Healthy Lungs, Hunter Medical Research Institute (HMRI), University of Newcastle, Callaghan, NSW 2308, Australia; dennis.thomas@newcastle.edu.au; 3Independent Researcher, Aberdeen AB32 6RU, UK; akmaclure@outlook.com (A.M.); katiemaclure@outlook.com (K.M.)

**Keywords:** smoking cessation, Saudi Arabia, systematic review

## Abstract

Literature on smoking in Saudi Arabia is extensive. However, studies capturing the attitudes of both smokers and non-smokers towards smoking cessation are few. A PRISMA-P protocol guided systematic searches in MEDLINE and CINAHL on MeSH terms (smoking cessation AND Saudi Arabia). Peer reviewed articles in English were included in the narrative analysis. Screening reduced the 152 articles identified to 15 and independent critical appraisal identified 10 final articles for review. Few adopted validated survey tools or mentioned the best practice to be followed. There was considerable variation in the prevalence of smoking reported (13.7–49.2%) and survey response rates (8.9–100%). There was a paucity of quality evidence but it is clear that the smoking pandemic is still resonant in Saudi Arabia. Despite support for education programs to prevent the uptake of smoking, policy-driven action to reduce environmental second-hand smoking, and provision of support for smoking cessation, more needs to be done.

## 1. Introduction

Smoking is acknowledged by the World Health Organization (WHO) as a global, long-term health issue linked to over 7 million deaths each year [1]. Furthermore, WHO states that, “tobacco is the only legal drug that kills many of its users when used exactly as intended by manufacturers”. With an estimated 1.2 million non-smokers dying from the effects of second-hand smoke, the overall annual death toll is over 8 million [1].

During the 2019–20 coronavirus pandemic, it was prescient that WHO reported that “although often associated with ill-health, disability and death from noncommunicable chronic diseases, tobacco smoking is also associated with an increased risk of death from communicable diseases” [1]. There is some evidence from early studies that smoking is a relevant factor in the likelihood of an individual contracting Covid-19, of being hospitalized as a result of Covid-19, and the severity with which Covid-19 is experienced. However, it should be noted that this is a living rapid review and has not yet been peer reviewed [2].

The key focus for both WHO and the Centre for Disease Control (CDC) is based on the WHO MPOWER strategy to-**M**onitor tobacco use and prevention policies; **P**rotect people from tobacco smoke; **O**ffer help to quit tobacco use; **W**arn about the dangers of tobacco; **E**nforce bans on tobacco advertising, promotion and sponsorship; and **R**aise taxes on tobacco [1,3]. The third item, “**O**ffer help to quit tobacco use”, acknowledges the difficulties smokers face in trying to give up highly addictive smoking habits, noting, “cessation support can more than double the chance of successfully quitting” [1].

In 2018, Saudi Arabia became one of 23 ‘best practice’ countries highlighted by WHO for offering tobacco dependence treatment [1]. Having implemented Executive Regulations on Anti-smoking Law issued by Royal Decree with the stated aim “to combat smoking by taking all necessary measures and steps at the state, community and individual levels, to reduce all types of smoking habit among individuals of all ages” [4]. The Law in Saudi Arabia defines smoking as “the use of tobacco and its products, such as, cigarettes, cigars, tobacco leaves, tobacco molasses or any other product containing tobacco, either through cigarettes, cigars, pipe, snus, hookah, or chewing tobacco, or any other form”. This was updated in 2018 to include e-cigarettes and e-hookah and was further confirmed in a WHO report in 2019 [4,5,6]. A strategic objective of Saudi Arabia’s National Transformation Program (2020) is the need to improve health with a greater focus on smoking cessation programs [7].

A recent literature review identified the prevalence and risks associated with smoking in Saudi Arabia. It found that the problem was reported in the mid-1990s, yet tobacco remained a readily available, legal product, with nicotine as its addictive ingredient [8]. The review described tobacco smoking in Saudi Arabia as, “a major and modifiable risk factor for cardiac (and other) diseases”. Furthermore, the review asserted that the, “majority of smokers were found motivated to quit smoking” recommending “strategic planning, designing tobacco control programs according to sex, age groups and education levels” [8]. However, the single author literature review with an unusual bullet-pointed Methods section that does not follow normal referencing convention, reports the 75 included studies before starting the Results section, and was superficial by necessity, presenting no new insights. The title placed the focus on risks, all of which were already well-known [8].

Determinants of outcomes for smokers attending smoking cessation clinics in Saudi Arabia were explored for over 20 years [9]. Several studies in Saudi Arabia focused on the views and attitudes of adolescents, medical college staff and students, university students, and healthcare professionals participating in surveys and interviews [9,10,11,12,13,14,15,16,17,18,19,20,21,22,23].

A study from 2016 investigated the socio-demographic factors, patterns of use, and cessation behavior associated with smoking in South-West Saudi Arabia [21]. It found “considerable variation in smoking prevalence” amongst participants (n = 1497), with 48% of those smoking (n = 736) having attempted more than once to stop smoking. The study also reported that married, university-educated men with a high income were more likely to smoke [21].

A large scale national survey (n = 10,735) in 2013 found “indicators of tobacco consumption in Saudi Arabia are better than most of the countries of the Middle East region and high-income countries, there are many potential areas for improvement” [22]. Although pre-dating the WHO MPOWER and CDC strategies, there is clear alignment in the survey recommendations and national smoking cessation actions [1,3,22,23].

By the end of 2018, the Ministry of Health in Saudi Arabia “aimed to have 450 smoking cessation clinics with 50 based in trucks, 100 mobile clinics and 300 fixed clinics” [6,23]. In 2020, and according to Saudi M.O.H official website, 47 registered smoking cessation clinics were available and providing care in Jeddah city, however, no details were accessible for mobile clinics. With such a commitment, there is a clear need to critically review and summarize the current evidence on the attitudes of both smokers and non-smokers towards smoking cessation, alongside reported outcomes.

The literature on smoking in Saudi Arabia is extensive. However, a summary of studies that captures the attitudes of both smokers and non-smokers towards smoking cessation needs to be further explored. Therefore, this systematic review asked the research question, ‘What are the attitudes of smokers and non-smokers towards smoking cessation in Saudi Arabia?’

## 2. Methods

### 2.1. Approach

This systematic review followed best practice in design and reporting, as recommended in the PRISMA Statement (PRISMA-P for protocol development, the PRISMA Flow Diagram and Reporting Checklist), the Centre for Review and Dissemination (CRD) at the University of York, and validated tools for critical appraisal and GRADE for quality of evidence [24,25,26]. A protocol was written and submitted to CRD Prospero prior to commencing the review.

### 2.2. Search Strategy

Searches were conducted in MEDLINE and CINAHL and Google Scholar, based on the MeSH search phrase (smoking cessation OR smoking cessation interventions OR quit smoking OR stop smoking) AND Saudi Arabia. Only peer reviewed articles published in English were included and no date or study type limitations were added; alerts were set up to notify the research team of newly published articles meeting the search terms. Electronic database searches were conducted by two members of the research team. Lists of titles returned had duplicates removed before independent screening and comparison by two members of the research team. Where there was any doubt based on the title and abstract, the article was taken forward to the next stage of screening. Full text articles of the remaining titles were retrieved, screened independently, and compared by two members of the research team. The reference lists of the final articles identified were searched for any additional articles.

### 2.3. Inclusion and Exclusion Criteria

This systematic review focused on (i) people in Saudi Arabia, whether reported as (ii) current or past smokers or non-smokers (in terms of PICO, the Population), who shared their (iii) attitudes towards smoking cessation (PICO Intervention). There were no population exclusion criteria (age, gender, education) other than not reporting the views of healthcare professionals. Smoking, in its many forms and definitions as described above (PICO Context, rather than a Comparator), and all aspects of smoking cessation (interventions, stopping smoking, quitting and giving up smoking; PICO Outcomes) were the focus for this systematic review.

### 2.4. Outcome Measures

The primary outcome of this systematic review was to report smokers’ and non-smokers’ attitudes towards smoking cessation, including, reasons for starting and not stopping smoking, reasons to quit smoking, attitudes towards smoke-free policy and second-hand smoking, intention to stop smoking and smoking cessation attempts, smoking cessation methods and supports.

Secondary outcome measures report general demographic characteristics of the smoking and non-smoking population and study specific information, including, smoking types, setting, participation, and recruitment, and age of starting smoking. These outcomes were reported before the primary outcome measure as a contextual basis.

### 2.5. Critical Appraisal

Quality assessment was conducted independently, then compared by two members of the research team, using a validated tool from the Centre for Evidence Based Medicine (CEBM). Risk of bias was assessed, based on GRADE criteria, to gauge the quality of evidence as one of four levels (High, Moderate, Low, or Very low) [26]. Studies that were deemed to be ‘very low’ quality of evidence were excluded after data extraction.

### 2.6. Data Extraction

The data extraction form included author (publication year), aim of the study and smoking focus, setting, methodology and methods, population, sample size and response rate, and key findings. This was independently conducted and confirmed by two members of the research team. For completeness, data extraction is presented for all studies regardless of GRADE outcome, with one table for included studies (high, moderate, and low GRADE) and a separate table for excluded studies (very low GRADE) [26].

### 2.7. Data Synthesis

The approach to data synthesis was dependent on the types of articles included and how the authors would report the outcomes. Statistical and clinical heterogeneity was considered for meta-analysis or narrative synthesis to best describe the findings.

### 2.8. Ethical Review

Ethical review was not required for this desk-based systematic review of the published literature.

## 3. Results

### 3.1. Search Findings

The articles identified in the database searches (n = 152) in February 2020 had duplicates removed (n = 20) in the referencing software (Zotero 5.0.82). Remaining articles (n = 132) were screened by title and abstract against the inclusion criteria, with exclusions noted (n = 93). The full text of the remaining articles (n = 39) were screened, with reasons for further exclusions recorded (not focused on Saudi Arabia n = 7; not focused on attitudes of non-healthcare professionals n = 19), reducing the final set of articles for review to thirteen. A further article was added in April 2020, following an alert of a new publication, with an additional article identified and added from the reference list of the new publication (n = 15). The screening and eligibility process was captured in the PRISMA Flow Diagram (Figure 1) [25], with the identified articles provided in Table 1 [9,10,13,15,21,27,28,29,30,31,32,33,34,35,36]. There was a noticeable gap in publications between 1996 [13] and 2014 [21], while 2016 was the most prolific [13,21,28,29,30]. Due to the heterogeneity of the findings in the included studies, a narrative synthesis without meta-analysis is presented.

### 3.2. Critical Appraisal

#### 3.2.1. Study Types

The fifteen articles were subject to independent critical appraisal by two members of the review team, before comparison and agreement on GRADE quality of evidence (Appendix A. Critical Appraisal) [26]. Most of the articles were quantitative and survey-based (n = 13/15), with one qualitative (focus groups plus interviews) study [33]. Although not standard practice, the single study type article was assessed alongside the surveys on the CEBM critical appraisal tool, for ease of comparison.

#### 3.2.2. Research Questions and Methods

While all addressed clearly focused research questions around smoking cessation in Saudi Arabia, with appropriate research methods, some did not clearly describe their recruitment approach, or search strategy in the case of the review article, which had the potential to introduce selection and reporting biases [13,17,27,33]. In the survey-based and qualitative studies, it was not always clear whether the participants were representative of the target population [13,21,27,28,33]. Only two of the survey-based articles provided clear sample size calculations [31,36]. However, most did provide both statistical significance and confidence intervals for the main results but some did not [13,31,35,36].

#### 3.2.3. Response Rates and Participation Levels

Response rates varied greatly, ranging from a poorly reported 8.9% [30] to 99.8% [21], while another inferred 100%, without providing any explanation [36]. Many of the quantitative studies did not report a response rate nor ethical approval (n = 9) [9,10,15,27,28,29,32,34,36]. Although the qualitative article showed high levels of participation in the focus groups and interviews, the recruitment approach was not fully reported, there were very few current or former smokers involved, and minimal data were reported [33].

#### 3.2.4. Use of Validated Tools and Guidelines

Only five of the 14 survey-based studies adopted or adapted validated data collection tools which may have impacted the reliability of the findings [9,10,29,30,31].

#### 3.2.5. Quality of Evidence

Overall, five studies were deemed ‘Very low’ [13,28,30,33,34] on the GRADE quality of evidence with eight ‘Low’ [9,10,15,21,27,33,35,36], only two ‘Moderate’ [30,31], and none ranked as ‘High’.

### 3.3. Data Extraction

A data extraction table was created for both the included studies and the excluded studies (Appendix A) to promote transparency of reporting. However, only the included studies (n = 10) are further reported and discussed in this review [9,10,15,21,27,29,31,32,35,36].

### 3.4. Secondary Outcome Measures for Context

#### 3.4.1. Smoking Types

The terms used to describe the types of smoking within the scope of the articles were reported by half as cigarette smoking (n = 5) [9,10,21,27,31], smoking tobacco (n = 3) [15,32,35], tobacco product consumption, including cigarettes and waterpipes (n = 1) [36], and one study on pre-operative smoking cessation used the term tobacco smoking but collected data on smoking of cigarettes, cigars, hookah, and chewable tobacco (n = 1) [29].

#### 3.4.2. Setting

Half of the studies were conducted in Riyadh (n = 5) [15,27,29,32,36], with single studies in each of Al Madinah [10], Buraydah [9], Jazan [21], Makkah, and one Saudi-wide study [31]. Smoking cessation clinics (n = 3) [9,32,35], more general public healthcare organizations (n = 3) [21,29,31] and educational establishments (n = 3) [10,15,36], each accounted for nearly a third of the studies, with a single study conducted on social media (n = 1) [27].

#### 3.4.3. Population and Recruitment

The target participants for the studies were described as smokers (n = 2) [9,32]. These included, Saudi nationals (n = 2) [20,27], university students and staff (n = 2) [14,36], school students currently smoking (n = 1) [10], in- and out-patients (n = 1) [29], administrative staff in a healthcare setting (n = 1) [31], and female smokers (n = 1) [35]. While all studies were described as cross-sectional and quantitative, half were randomly sampled (n = 5) [9,15,21,31,36], others included three-stage cluster sampling (n = 1) [27], convenience sampling (n = 1) [29], or were lacking in recruitment details (n = 3) [10,32,35].

#### 3.4.4. Prevalence of Smoking

One study reported a prevalence of smoking at an exceptionally high level of 49.2% (n = 736) of those attending a public healthcare center [21]-not specifically for smoking cessation-while another showed university staff smoking prevalence was 36.8% (n = 153) [15], 30.3% (n = 61) in administrative staff [31], while in the social media study of the general population it was 39% (n = 802) [27].

#### 3.4.5. Age of Starting Smoking

Several of the studies collected data on age of initiating smoking. The most notable finding was amongst intermediate and secondary school students, some of whom indicated they first tried smoking under 10 years old, 5.3% (n = 47), with 66.8% (n = 205) aged between 10 and 15 years [10]. Another study confirmed 65.7% (n = 482) started smoking under the age of 18 years [21]. A further study of women attending smoking cessation clinics reported 8.3% (n = 181) were aged between 10 and 15 years old at the time of attending the clinic, with a further 5% (n = 110) aged between 16 and 20 years [35].

### 3.5. Primary Outcome Measures on Smoking Cessation

#### 3.5.1. Reasons for Starting and Not Stopping Smoking

There was commonality around the reasons for starting to smoke, often citing family members (74.7%, n = 550; 8.3%, n = 181) or friends (31.5%, n = 232; 31.1%, n = 682) as smokers [21,35]. Some participants indicated that smoking relieved boredom, or was entertaining (72.7%, n = 535), fun (45.2%, n = 79), or a social activity [21,36]. Others started smoking to reduce stress or tension (85.3%, n = 628; 5.7%, n = 125; 33.3%, n = 58) or viewed smoking as attractive (3.0%, n = 5) [21,35,36]. Reasons for not stopping were similar-peer pressure (18.5%, n = 405), fear of failure (11.2%, n = 245), and fear of mood changes (28%, n = 613), all cited in an article on female smoking [35].

#### 3.5.2. Reasons to Quit Smoking

The same article on female smoking cessation gave multiple reasons for trying to stop smoking, which were similarly reported amongst school students [35]. These included health concerns (45.5%, n = 996; 50.5%, n = 111), familial pressure (20.4%, n = 446; 16.4%, n = 36), religious beliefs (18.1%, n = 397; 14.5%, n = 32), or for a better life (14.3%, n = 314) [35]. Altumairi (2014) reported large numbers of non-smokers (students 98.2%, n = 3119; faculty 99.8%, n = 558; staff 98.1%, n = 258) who agreed that “smoking can cause harm for myself” [15]. Smokers were also largely in agreement that “smoking can cause harm for myself” (students 94.5%, n = 380; faculty 94.7%, n = 36; staff 98.7%, n = 151) and this pseudo measure of smoking-related harm and health knowledge was reflected in other studies, including second-hand smoking [10,15,36]. Hajjar et al. (2016) reported on patients’ attitudes towards pre-operative smoking cessation, found that 66.7% (n = 72) of surgery patients and 73.1% (n = 502) of non-surgery patients “were unaware of the harmful effects of smoking” [29]. Some wanted to quit smoking to save money (50.5%, n = 111; 14.7%, n = 323;) [10,35]. A narrow majority of smokers (56%, n = 454) thought cigarette price was expensive while non-smokers (57.6%, n = 720) thought cigarettes were cheap [27]. However, a 2014 study on the potential impact of cigarette pricing found the price would need to treble “to observe a significant reduction in cigarette consumption” [27].

#### 3.5.3. Attitudes towards Smoke Free Policy and Second-Hand Smoking

Altumairi (2014) also reported that both smokers (students 61.4%, n = 247; faculty 60.5%, n = 23) and non-smokers (students 86.5%, n = 2747; faculty 71.0%, n = 397; staff 83.6%, n = 220) were in agreement that “smoking should be banned in all public places” [15]. However, only one in five staff who smoked were in agreement (staff 20.9%, n = 32) [15]. Support was even higher for ”our campus should be completely smoke free” from all groups of participants. Al-Zalabani et al. (2015) found lower levels of support for a “smoking ban in public places” amongst school student smokers who intended to quit (64.5%, n = 142) and those who had no intention to quit (47.1%, n = 41) [10]. Amin et al. (2020) reported on environmental exposure to second-hand smoke [36]. Just over half of the healthcare students reported “tobacco product use by a family member” (52.6%, n = 669) with close to a third (31.7%, n = 403) “sitting with a family member who smokes” and a much higher proportion reporting “sitting with friends who smoke” (42.5%, n = 541) was their usual social practice [36].

#### 3.5.4. Intention to Quit and Quit Attempts

Salih and Farghaly (1996) reported that 38.3% (n = 125) of participants attending an anti-smoking center were described as ‘quitters’ at time of follow-up [9]. Further, 96% of ‘quitters’ had a past history of more than one attempt to stop, with 30.4% making three or more attempts. Additionally, 35% of ‘quitters’ had previously stopped for 3 months or longer [9]. Altumairi (2014) compared whether current smokers (n = 593) amongst university students (62.0%), faculty (80.0%) and staff (28.9%) had made an attempt to quit [15]. Likewise, Al-Zalabani et al. (2015) found school students (70.5%, n = 155) with an intention to quit had a history of attempts to quit within the last year [10]. Amongst those school students, intention to quit was higher amongst those aged 17 or older (53.6%, n = 118), compared to those aged 13–16 (43.2%, n = 95) or those aged 12 and under (3.2%, n = 7) [10]. Of note is Al-Zalabani et al.’s finding that participants who were willing to quit smoking were more likely to have non-smoking parents and friends [10]. The majority (56.8%; n = 417) tried to quit smoking in a study by Abdelwahab et al. (2016), with 8.3% (n = 61) making more than four attempts [21]. In the same study, when asked “will you quit smoking in the future?”, most responded “possibly yes” (44.4%, n = 326), with a quarter stating “surely yes” (24.8%, n = 182) [21]. Amongst non-healthcare professionals working as administrators in a healthcare setting (n = 697), Mahdi et al. (2018) found 19.9% (n = 139) were current smokers, of whom 64.9% (n = 85) “want to stop smoking now” with more than two-thirds having “tried to stop during the past year” (70.2%, n = 92) [31]. Mahdi et al. (2018) also found that 75.0% (n = 99) thought “healthcare workers who smoke are less likely to advise patients to stop smoking” [31]. In the only study to focus solely on women, Al-Nimr et al. (2020) reported 26.3% (n = 575) had “previous attempts at quitting” [35].

#### 3.5.5. Smoking Cessation Methods and Supports

Abdelwahab et al. (2016), found a smoking prevalence of 49% (n = 736) and asked participants for their ‘perception on the most successful cessation program’ [21]. Respondents perceived ‘school awareness program’ (88.6%, n = 650), ‘nicotine mixed gum (NRT)’ (70.9%, n = 521), ‘TV’ (78.5%, n = 577), and ‘radio’ (74.4%, n = 547) awareness campaigns to be the most successful [21]. The most popular supports for smoking cessation were noted by Mahdi et al.’s (2018) study, which reported administrative staff attitudes that “healthcare organizations should establish smoking cessation clinics” (86.5%, n = 109) and also that “healthcare workers should get specific training on cessation techniques” (84.8%, n = 106), as they reported “healthcare workers have a role in giving advice about smoking cessation to patients” (88.9%, n = 112) [31]. The authors also raised that “patients’ chances of quitting smoking increase if a healthcare worker advises him/her” (76.6%, n = 95) [31].

In the only study to focus on the use of social media-based support groups, Onezi et al. (2018) found 44.7% (n = 227) “would recommend participating in a social media support group to prevent smoking relapse” [32]. The participants using WhatsApp or Twitter reported they had reduced their smoking frequency (40.6%, n = 210) and were “satisfied that social media support groups help to prevent relapse” (39.5%, n = 204) [32].

## 4. Discussion

### 4.1. Context and Meaning

This systematic review set out to answer the question “What are smokers and non-smokers attitudes towards smoking cessation in Saudi Arabia?” First, there is a scarcity of quality articles on which to confidently base any answer. Although no robust conclusions could be drawn, based on the current evidence, this review gives an insight into peoples’ attitudes towards tobacco smoking, the problems associated with identifying smokers, providing smoking cessation interventions, some useful tips on intervention delivery (Radio, TV, and social media) and some potential interventions (school awareness program, price increase, NRT). This review also highlights the prevalence of smoking among young people and administrative workers in healthcare settings, and the need for better designed, adequately powered surveys.

As recently as 2017, WHO estimated male smoking levels at 31.2%, with females at 2.1% in Saudi Arabia [37]. These figures are often presented as a combined percentage, which might be disguising higher levels for males. Likewise, the recent study by Al-Nimr et al. (2020) recruited females from first time attendees at 18 smoking cessation clinics across Saudi Arabia [35]. Over the 3-year period, 3000 women were invited to participate, challenging previously reported female smoking levels and difficulty in recruiting the female population for research, which has often been explained to be a result of culture, as reported elsewhere [10,15,21,36,37].

With only 10 papers included in the review, two of which achieved a quality of evidence GRADE of at least moderate, the evidence base to support particular approaches to smoking cessation is lacking [26]. Noticeable too is the gap between the study included from 1996 [10], through to two studies in 2014 [15,27], before there was any further research activity. Given the Saudi Ministry of Health Royal Decree (2015) [4] and Tobacco Control Law (2018) [5], it is to be hoped that research evaluating the measures taken to promote smoking cessation in the country would be forthcoming and some is indeed in press [38]. Certainly, the WHO Report on the ‘Global Tobacco Epidemic: Offer help to quit tobacco use’ (2019) should serve to stop any complacency that smoking cessation is not still a major, global (including Saudi Arabian) public health issue [1]. During the Covid-19 pandemic, all outdoor smoking in public spaces was banned; time will tell whether the Saudi general public will be receptive to maintaining this ban.

Of particular concern is the continued uptake of smoking amongst young people [13,15,21,37,38]. Therefore, while it is laudable for studies to recommend that education programs should focus on preventing school students from starting to smoke, clearly the message is not having the desired impact. Additionally, of concern is the level of smoking amongst current administrative workers in healthcare settings and healthcare students, and how this impacts their ability to encourage patients to quit smoking in the future [31,33,36]. The live review ongoing during the Covid-19 pandemic might shed light on factors relating to smokers’ risk of catching the virus, the severity of disease experienced by smokers relative to non-smokers, and the risk of being hospitalized [2].

With no new insight into why people start smoking, and why they subsequently find it difficult to stop, social media-based support groups might offer variation on the usual approaches to smoking cessation [32]. The need to treble the price of smoking before any noticeable impact on smoking cessation will be disheartening news for healthcare policy advisors [27]. Therefore, while university campuses, hospitals, and other public places might be smoke free, second-hand smoking with friends and family remains problematic [10,13,35,36]. A recent study by Itumalla and Aldhmadi, currently in press, reports on “initiatives taken from June 2017 to April 2019 by the Saudi government to combat tobacco use, including value-added tax on tobacco, antismoking campaigns, antismoking clinics, mobile apps and other initiatives” [38]. They conclude, and this systematic review supports their view, “that the Government should evaluate the impact of these initiatives on tobacco control in Saudi Arabia” [37].

### 4.2. Strengths and Limitations

This systematic review followed best practice throughout, starting with an experienced, multidisciplinary review team, with the application of recognized guidelines, use of independent researchers for screening, critical appraisal, and data extraction, based on established tools [23,24,25,39,40,41]. Although the protocol was submitted to CRD Prospero, the rapid progress in conducting the review meant it was not in the end registered. Despite our attempts, additional databases could potentially have been searched (for example, TRIP, ScienceDirect, PubMed), so we cannot rule out the possibility of missing some important studies. There is always a potential risk of publication bias too.

## 5. Conclusions

In conclusion, the paucity of quality evidence on which to base any recommendations means that the findings of this systematic review should be acted upon with caution. What is clear is that the smoking pandemic is still resonant in Saudi Arabia, and that research is not keeping pace [1,3,4,38]. Despite strong Ministry of Health support for education programs that try to prevent the uptake of smoking, policy-driven action to reduce environmental second-hand smoking, and provision of support for smoking cessation [4,5,7], more needs to be done and further research needs to be undertaken to evaluate outcomes.

## Figures and Tables

**Figure 1 ijerph-17-08194-f001:**
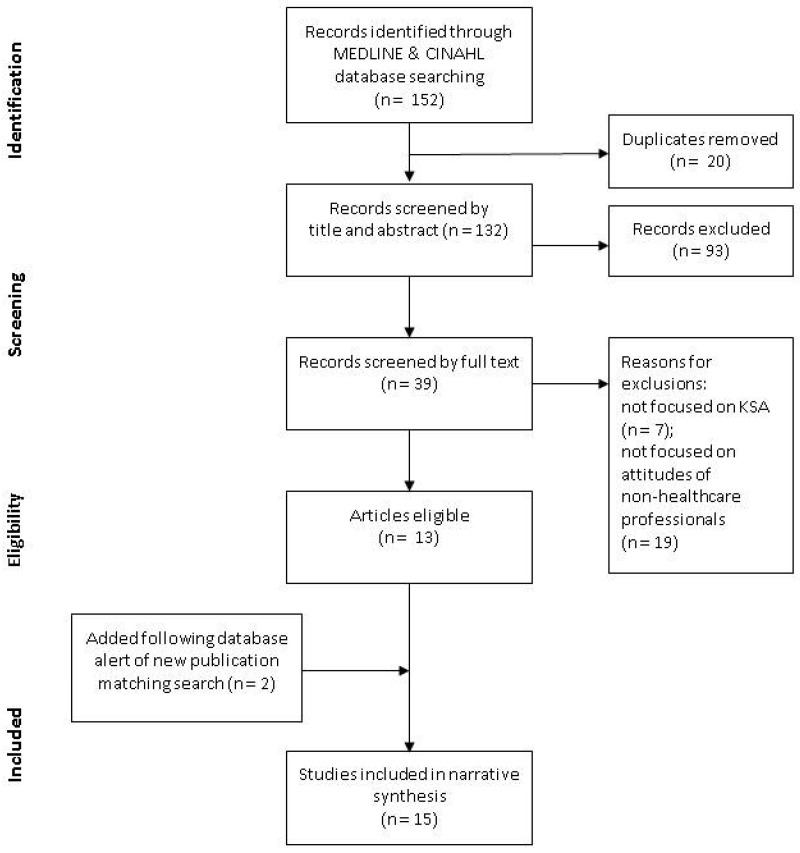
PRISMA Flow Diagram [25].

**Table 1 ijerph-17-08194-t001:** Articles identified in the review.

Year of Publication	Authors	Title	Journal
1996 [9]	Salih MA & Farghaly AA	Determinants of outcome among smokers in a smoking cessation program	Journal of Family & Community Medicine
2014 [15]	Almutairi KM	Attitudes of Students and Employees Towards the Implementation of a Totally Smoke Free University Campus Policy at King Saud University in Saudi Arabia: A Cross Sectional Baseline Study on Smoking Behavior Following the Implementation of Policy	Journal of Community Health
2014 [27]	Al-Mohrej OA, AlTraif SI, Tamim HM, et al.	Will any future increase in cigarette price reduce smoking in Saudi Arabia?	Annals of Thoracic Medicine
2015 [10]	Al-Zalabani AH, Abdallah AR, Alqabshawi RI	Intention to Quit Smoking among Intermediate and Secondary School Students in Saudi Arabia	Asian Pacific Journal of Cancer Prevention
2016 [13]	Baig M, Bakarman MA, Gazzaz ZJ, et al.	Reasons and Motivations for Cigarette Smoking and Barriers against Quitting Among a Sample of Young People in Jeddah, Saudi Arabia	Asian Pacific Journal of Cancer Prevention
2016 [21]	Abdelwahab SI, El-Setohy M, Alsharqi A, et al.	Patterns of Use, Cessation Behavior and Socio-Demographic Factors Associated with Smoking in Saudi Arabia: a Cross- Sectional Multi-Step Study	Asian Pacific Journal of Cancer Prevention
2016 [28]	Alyamani MJ, Alkriadees YA, Alkriadees KA, et al.	Determinants and Predictors of Smoking Cessation among Undergraduate and Graduate Medical Students: a Cross-Sectional Study in a Private Medical College	International Journal of Advanced Research
2016 [29]	Hajjar WM, Al-Nassar SA, Alahmadi RM, et al.	Behavior, knowledge, and attitude of surgeons and patients toward preoperative smoking cessation	Annals of Thoracic Medicine
2016 [30]	Almogbel YS, Abughosh SM, Almeman AA, et al.	Factors associated with the willingness to quit smoking among a cohort of university students in the KSA	Journal of Taibah University Medical Sciences
2018 [31]	Mahdi HA, Elmorsy SA, Melebari LA, et al.	Prevalence and intensity of smoking among healthcare workers and their attitude and behavior towards smoking cessation in the western region of Saudi Arabia: A Cross-sectional study	Tobacco Prevention & Cessation
2018 [32]	Onezi HA, Khalifa M, El-Metwally A, et al.	The impact of social media-based support groups on smoking relapse prevention in Saudi Arabia	Computer Methods and Programs in Biomedicine
2018 [33]	Jradi H, Saddik B	Graphic warnings and text warning labels on cigarette packages in Riyadh Kingdom of Saudi Arabia: Awareness and perceptions	Annals of Thoracic Medicine
2019 [34]	Alqurashi AA, Alluhaybi HF, Al-raddadi R	Smoking Cessation Outcomes and Predictors Among Individuals Enrolled in the Anti-Tobacco Program in Jeddah 2018	Indo American Journal of Pharmaceutical Sciences
2020 [35]	Al-Nimr YM, Farhat G, Alwadey A	Factors Affecting Smoking Initiation and Cessation Among Saudi Women Attending Smoking Cessation Clinics	Sultan Qaboos University Medical Journal
2020 [36]	Amin HS, Alomair AN, Alhammad AH, et al.	Tobacco consumption and environmental exposure among healthcare students in King Saud University in Riyadh	Journal of Family Medicine and Primary Care

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
