# Peer review of "Smokers’ and Non-Smokers’ Attitudes towards Smoking Cessation in Saudi Arabia: A Systematic Review"

_ijerph, 2020, doi:10.3390/ijerph17218194_

Round 1
Reviewer 1 Report
The manuscript is an in-depth literature review of smokers and non-smokers attitude toward smoking cessation.
INTRODUCTION:
- There are several undefined acronyms
- The Statement about COVID-19 seems irrelevant to the study. I recommend removing it
METHOD: well written and developed
DISCUSSION: Well developed and of significant interest to potential readers
Author Response
Many thanks for your time and expertise in reviewing our article and your kind comments. In response to your points listed:
- Please can you identify the undefined acronyms as we are possibly too close to the manuscript to see which we have overlooked to expand.
- The SR was conducted during the Covid-19 pandemic which is still ongoing. Therefore, we did feel it was of emerging and ongoing relevance. Given our topic area, not to mention covid-19 may have been viewed by some as an oversight.
Reviewer 2 Report
I think the topic of the article is very interesting.
On the other hand, I have several major concerns about this work:
1.- The authors declared to follow the PRISMA-P guidelines. However, the article is not structured according to it. I suggest the authors follow the checklist in table 3 reference [24] or table 1 reference [25].
2.- Lines 31-33: I suggest changing the reference [2] for another peer-reviewed work.
3.- Line 35: Where is the location of the Centre for Disease Control?
4.- Line 35: Please provide some reference for WHO MPOWER.
5.- Lines 35-39: I do not understand what is the idea of this paragraph. I think it needs to be rewritten. Why the authors say "third item"?
6.- Line 43: A single quotation mark is missing at the end of the sentence.
7.- Lines 51-55: The reference [8] presents a similar work, with a similar objective. However, that work is not adequately analyzed in this work. Provide a better analysis (a comparative one might be fine).
8.- Line 88: Why did the authors choose those scientific databases? Please provide a comprehensive justification and indicate why you did not choose others such as Springer, ScienceDirect, Emerald, PubMed.
9.- Line 91: I do not understand the sentence "alerts were set up".
10.- Lines 98-103: I think the redaction of this paragraph is a bit confuse. I am not clear where the inclusion and the exclusion criteria begin or end. Furthermore, and to the best of my knowledge, some important inclusion/exclusion criteria are missing:
* article language (English)
* year of publication (range)
* origen of the publication (journal, conference, other)
11.- Line 99: The sentence "who have shared their attitudes towards smoking cessation" is difficult to decode as an inclusion criterion. Can you change the redaction of it?
12.- Section 2.4: According to PRISMA the authors must present research questions and PICOS, PICO or similar.
13.- Line 126-127: I do not understand the sentence "The approach to data synthesis was dependent on the types of articles included and how the authors would report the outcomes". Please provide a comprehensive explanation.
14.- Line 127-128: I can not find in this article the "meta-analysis or narrative synthesis". Can you indicate where they are?
15.- Section 3:
a) The way the results are displayed must change. They are presented article by article, showing specific data from them. However, the information is not grouped by topic. Readers of a SLR not only need raw article information from the articles but also summary information. The authors do not provide any graph that summarizes the information obtained.
b) Please cite articles immediately where applicable. It is wear to see sentence as "Overall, six of the studies were deemed ‘Very low’ on the GRADE quality of evidence with eight ‘Low’, only two ‘Moderate’ and none ranked as ‘High’ [13,16,19,28,30,34], [9,10,15,21,27,33,35,36], [30,31].", where groups of references are separated by commas (Lines 176-177).
16.- Line 36: Eliminate sentence "attitudes towards smoking cessation AND Saudi Arabia".
17.- Line 141: Diagram25 ---> Diagram [25].
17.- Lines 141-142: Please eliminate sentence "There is a noticeable gap in publications between 1996 and 2014 with 2016 the most prolific [13,21,28–30]".
18.- Figure 1: Please eliminate the PRISMA logo, eliminate the reference to PLoS Med, and cite it properly in the text.
19.- Table 1: Reference [16] must be removed because reviews are not included in a systematic literature review.
20.- Discussion: This section have to be rewritten because it is not a discussion. A discussion is a critical reflexion that the authors do in relation to their findings. They relate and compare their results with the results of other authors/works.
21. Line 299: Cite properly "Al-Nimr et al (2020)"
22.- Line 305-306. Provide a comprehensive explanation about it.
23.- Conclusion: This section must be rewritten. Sentence in lines 333 to 336 are part of the discussion. The sentence in lines 336 to 336 is not a conclusion because it does not follow from the results of this work.
Author Response
Many thanks and we especially thank you for your time and expertise in peer reviewing our SR. We have tried to address each of your points as follows.
On the other hand, I have several major concerns about this work:
1.- The authors declared to follow the PRISMA-P guidelines. However, the article is not structured according to it. I suggest the authors follow the checklist in table 3 reference [24] or table 1 reference [25].
There may be some confusion here. The PRISMA-P guidelines [24] help researchers write a SR protocol in advance of conducting a SR which we did but was not part of the submitted SR. The SR article was structured and reported in order of the PRISMA Checklist [25]. The checklist which was provided as a supplementary table has the page number added to help locate each item in the SR. I hope that helps make sense of our approach.
2.- Lines 31-33: I suggest changing the reference [2] for another peer-reviewed work.
Please can we suggest you look again at this living review which is now on version 9 with over 200 references and growing number of citations? This is undergoing open peer review with updated reference:
David Simons, Lion Shahab, Jamie Brown, Olga Perski. (2020). The association of smoking status with SARS-CoV-2 infection, hospitalisation and mortality from COVID-19: A living rapid evidence review with Bayesian meta-analyses (version 8). Qeios. doi:10.32388/UJR2AW.9. [Accessed 24 October 2020]
3.- Line 35: Where is the location of the Centre for Disease Control?
This information is available at reference [3] which appears at the end of the sentence.
4.- Line 35: Please provide some reference for WHO MPOWER.
We have double-checked that reference [1] is appropriate for WHO MPOWER.
5.- Lines 35-39: I do not understand what is the idea of this paragraph. I think it needs to be rewritten. Why the authors say "third item"?
The MPOWER acronym third letter ‘O’ is ‘Offer help to quit tobacco use’ this paragraph relates the global context, extent and continuing issue of smoking and the tools from WHO and CDC. We wondered if perhaps it was not clear that MPOWER is an acronym so have made the first letter in bold of each definition. We hope that helps the reader.
6.- Line 43: A single quotation mark is missing at the end of the sentence.
Thank you for highlighting – amended line 45.
7.- Lines 51-55: The reference [8] presents a similar work, with a similar objective. However, that work is not adequately analyzed in this work. Provide a better analysis (a comparative one might be fine).
We have extended our commentary on reference [8].
8.- Line 88: Why did the authors choose those scientific databases? Please provide a comprehensive justification and indicate why you did not choose others such as Springer, ScienceDirect, Emerald, PubMed.
We have found on past experience these provide comprehensive coverage of the peer reviewed health-related literature in KSA. Of course more database search tools could have been used. Ultimately this results in greater duplication of the same articles from different sources. The following has been added to the Limitations:
‘…additional databases could have been searched (for example TRIP, ScienceDirect, PubMed)…’
9.- Line 91: I do not understand the sentence "alerts were set up".
The statement has been extended ‘alerts were set up to notify the research team of newly published articles meeting the search terms’ to aid understanding.
10.- Lines 98-103: I think the redaction of this paragraph is a bit confuse. I am not clear where the inclusion and the exclusion criteria begin or end. Furthermore, and to the best of my knowledge, some important inclusion/exclusion criteria are missing:
* article language (English)
* year of publication (range)
* origen of the publication (journal, conference, other)
This is covered in the search strategy lines 90-91 which we have highlighted.
11.- Line 99: The sentence "who have shared their attitudes towards smoking cessation" is difficult to decode as an inclusion criterion. Can you change the redaction of it?
We have added numbering to aid the reader.
This systematic review focused on (i) people in Saudi Arabia, whether reported as (ii) current or past smokers or non-smokers, who have shared their (iii) attitudes towards smoking cessation.
12.- Section 2.4: According to PRISMA the authors must present research questions and PICOS, PICO or similar.
The words ‘the research question’ have been added after the stated aim in line 78.
We have made the PICO more explicit as follows,
This systematic review focused on (i) people in Saudi Arabia, whether reported as (ii) current or past smokers or non-smokers (in terms of PICO, the Population), who have shared their (iii) attitudes towards smoking cessation (PICO Intervention). There were no population exclusion criteria (age, gender, education) other than not reporting the views of healthcare professionals. Smoking, in its' many forms and definitions as described above (PICO Context, rather than a Comparator), and all aspects of smoking cessation (interventions, stopping smoking, quitting and giving up smoking; PICO Outcomes) were the focus for this systematic review.
13.- Line 126-127: I do not understand the sentence "The approach to data synthesis was dependent on the types of articles included and how the authors would report the outcomes". Please provide a comprehensive explanation.
Only once the final set of articles for the review were identified would we know the type of data for the synthesis and subsequent reporting. It is not possible to know in advance whether a meta-analysis is possible or narrative synthesis more appropriate.
14.- Line 127-128: I can not find in this article the "meta-analysis or narrative synthesis". Can you indicate where they are?
We have added the following statement to the 3.1:
Due to the heterogeneity of the findings in the included studies a narrative synthesis without meta-analysis is presented.
15.- Section 3:
a) The way the results are displayed must change. They are presented article by article, showing specific data from them. However, the information is not grouped by topic. Readers of a SLR not only need raw article information from the articles but also summary information. The authors do not provide any graph that summarizes the information obtained.
We presented the Results grouped under subheadings based on the pre-determined and stated primary and secondary outcomes. We did not find any Results suitably homogenous for presenting in graphical form.
- b) Please cite articles immediately where applicable. It is wear to see sentence as "Overall, six of the studies were deemed ‘Very low’ on the GRADE quality of evidence with eight ‘Low’, only two ‘Moderate’ and none ranked as ‘High’ [13,16,19,28,30,34], [9,10,15,21,27,33,35,36], [30,31].", where groups of references are separated by commas (Lines 176-177).
Amended as requested.
16.- Line 136: Eliminate sentence "attitudes towards smoking cessation AND Saudi Arabia".
Statement deleted as requested.
17.- Line 141: Diagram25 ---> Diagram [25].
Typo corrected.
17.- Lines 141-142: Please eliminate sentence "There is a noticeable gap in publications between 1996 and 2014 with 2016 the most prolific [13,21,28–30]".
We believe this is a matter of note so prefer to retain but have amended to:
There is a noticeable gap in publications between 1996 [13] and 2014 [28] while 2016 was the most prolific [28–30].
18.- Figure 1: Please eliminate the PRISMA logo, eliminate the reference to PLoS Med, and cite it properly in the text.
Thank you, we were not aware PRISMA had updated their permission statement requirements. All amended.
19.- Table 1: Reference [16] must be removed because reviews are not included in a systematic literature review.
We stated all study types would be included but respect your opinion so have amended throughout the abstract, main text, tables and figure.
20.- Discussion: This section have to be rewritten because it is not a discussion. A discussion is a critical reflexion that the authors do in relation to their findings. They relate and compare their results with the results of other authors/works.
Amendments have been made and references added.
- Line 299: Cite properly "Al-Nimr et al (2020)"
We cannot see what is to be corrected here but noticed incorrectly cited as 2019 earlier in text.
22.- Line 305-306. Provide a comprehensive explanation about it.
We are unsure what more you are suggesting as the point is further explored in the rest of that paragraph.
23.- Conclusion: This section must be rewritten. Sentence in lines 333 to 336 are part of the discussion. The sentence in lines 336 to 336 is not a conclusion because it does not follow from the results of this work.
The sentence has been relocated to the Discussion and the section amended.
Reviewer 3 Report
The central theme of the review is of interest to advance knowledge of smoking cessation. The review is understandable and structured.
Observations:
The review could have been carried out in more databases and clinical search engine (for instance TRIP), avoiding Google Scholar as it is a non-exhaustive publication database.
The authors noted: "There were no population 100 exclusion criteria (age, gender, education) other than not reporting the views of healthcare professionals". My question is: Were health professionals included as the study population? It's understood that their opinions do not. So, the authors can explain the non-inclusion of the following study: Awan KH, Hammam MK, Warnakulasuriya S. Knowledge and attitude of tobacco use and cessation among dental professionals. Saudi Dent J. 2015 Apr;27(2):99-104. doi: 10.1016/j.sdentj.2014.11.004. If this manuscript is among those excluded, could the authors state the reasons?
It would be of interest to add a figure with the articles found and excluded in each database used.
Author Response
Many thanks for your time and shared expertise in peer reviewing our article. We welcome the opportunity to address the points you have raised:
- Thank you for the suggestion. I have not used TRIP database before but will do so going forward. However, we do feel we were able to capture the relevant articles. Google Scholar acted as a double check but I’ve checked back on our spreadsheets and no additional articles were identified in that way. The PRISMA Flow diagram has been updated.
- No, healthcare professionals were not included in the study population. We have highlighted the part of the manuscript in which we state that healthcare professionals views are not included also in the text you copied from the manuscript above. We are aware there is an equally large body of literature on the views of health care professionals so took the decision to separate out the two populations. We did this because the research questions asked of each population would be quite different. There is the potential for another SR on their views and yet another comparing the views of the two populations but to do all in one SR would be too much content for a single article potentially leading to only superficial findings.
- As per response in 2 above, it was not our intention to include healthcare professionals’ views but I can confirm the article by Awan et al was identified during the search.
- We have taken on board your comments in item 1 above adding MEDLINE and CINAHL to the PRISMA Flow Diagram. We did not keep track of how many from each. Producing such a figure would present further difficulties where articles were duplicated across both databases so which database do we attribute articles which remained in the SR. I hope you agree that keeping to the PRISMA accepted standard representation is appropriate.
Round 2
Reviewer 2 Report
Many thanks to the authors for considering my suggestions.
I think the article has improved substantially. For my part, no further corrections are necessary.